# Effect of Rural Human Settlement Environment around Nature Reserves on Farmers’ Well-Being: A Field Survey Based on 1002 Farmer Households around Six Nature Reserves in China

**DOI:** 10.3390/ijerph19116447

**Published:** 2022-05-26

**Authors:** Tingting Zhang, Dan He, Tian Kuang, Ke Chen

**Affiliations:** 1College of Economics and Management, Shenyang Agricultural University, Shenyang 110866, China; hedanonline@sina.com (D.H.); kuangtian@syau.edu.cn (T.K.); 2College of Applied Technology, Shenyang University, Shenyang 110044, China

**Keywords:** nature reserves, human settlement environment, well-being, farmers

## Abstract

Numerous countries actively consider the human settlement environment and have implemented rural governance strategies to ameliorate the living conditions of rural dwellers. The construction of a rural human settlement environment is an important goal of China’s rural revitalization strategy and improving farmers’ well-being is a key element of China’s policies on agriculture, farmers, and villages. However, whether a rural human settlement environment enhances farmers’ well-being remains untested. By adopting the method of random stratified sampling, this study investigated 1002 farmers inside and outside six nature reserves in Liaoning, China. OLS and ordered probit regression models were used to assess the impact on the well-being and the satisfaction of farmers with their settlement environment around nature reserves from three aspects: the natural ecological environment, the hardware facility environment, and the daily governance environment. The results of this study proved that the construction of a human settlement environment can significantly boost the well-being of farmers. Moreover, the satisfaction towards the natural ecological environment, hardware facility environment, and daily governance environment exerts a substantial impact on the well-being at the significance level of 1%, with a positive sign, showing a stable enhancement role. Among them, the satisfaction with the hardware facility environment was the most essential for improving happiness, with a coefficient of 0.126. A heterogeneity analysis suggests that the positive effect of satisfaction with the human settlement environment on farmers’ well-being within nature reserves was more significant in the natural ecological environment, with a coefficient of 0.244; the hardware facility environment had the greatest positive effect on the well-being of farmers outside nature reserves, with a coefficient of 0.224; and the daily governance environment had a greater enhancing effect on the well-being of farmers both inside and outside nature reserves. Based on these results, it is recommended that governments encourage farmers around nature reserves to participate in wildlife accident insurance, strengthen ecological environmental protection, and enhance the hardware facility environment. Furthermore, local governments should disseminate knowledge of human settlement management to farmers and improve the efficiency of human settlement environment management at the grassroots level. Finally, governments should prioritize human settlement environment development and identify the farmers’ needs of human settlement environment to enhance their well-being.

## 1. Introduction

Subjective well-being is a crucial measure of social development and national governance and has become increasingly important in fields such as public policy [1]. In 2013, the French Commission for the Measurement of Economic Performance and Social Progress reported that economic indicators such as gross domestic product are insufficient to measure social progress and that various governments globally are now increasingly using subjective well-being to assess policy effects and monitor human progress [2]. The World Happiness Report published by the United Nations ranked 156 countries on their level of national well-being, confirming that subjective well-being is an important indicator of social development [3]. China’s overall development level has crossed the stage of survival and is moving towards a stage of development, which is the stage where people’s requirements for higher well-being are met. People’s aspiration for a better life is becoming increasingly strong. The report of the 19th National Congress of the Communist Party of China, (CPC) stated that “we should improve the public service system and guarantee the basic livelihood of the people to continuously meet their growing needs for a better life. Ultimately, the people’s sense of access, well-being, and security will be more fulfilled and enhanced.” The aim to enhance public well-being reflects that in China’s new era, the existing social contradictions are changing, as improved well-being is the main intention and purpose of the country’s developmental efforts.

With a population of over 500 million, farmers’ well-being is a great concern in China. Income is the economic basis of people’s well-being. The average growth rate of farmers’ income in China is 11.92% [4]. Farmers’ well-being brought about by a growth in wealth decreases as their income level increases [5,6]. Additionally, income inequality between urban and rural areas erodes farmers’ well-being brought about by economic growth [7]. the overall well-being of Chinese farmers has not yet reached a high level [8]. Thus, breaking out of the “well-being stagnation” dilemma requires shifting focus from improving farmers’ economic situation to actively exploring other aspects that affect their well-being. Among these, the farmers’ human settlement environment is crucial. In 2018, the General Office of the CPC Central Committee and the General Office of the State Council issued the Three-Year Action Plan for the Improvement of the Rural human settlement Environment, which aimed to improve the rural human settlement environment and build a beautiful and livable countryside. China’s No. 1 Central Document for 2022 stated that “We should take account of the actual needs of farmers and implement successively the five-year action to improve and upgrade the rural human settlement environment.” The three-year action program for the construction of human settlement environments has now been concluded, while the five-year action plan has only just begun. An improved human settlement environment is expected to have a profound positive effect on farmers’ well-being.

So far, there is no uniform definition of rural human settlement environment. However, there is general agreement that the rural human settlement environment is a phenomenon that involves the interaction of natural, economic, social, and cultural elements of the environment arising from farmers’ lives and activities and comprising an organic combination of material and nonmaterial elements [9,10]. When constructing human settlement environment index systems and evaluating the effects of improvement efforts, scholars mainly focus on ecology, production, and living [11,12]; “hard environment” factors such as the level of public service facilities; “soft environment” factors such as the level of social services [13,14]; and the use of household toilets and domestic waste and sewage treatment [15,16], from the macro and meso perspectives. The construction of a rural human settlement environment is a governance service provided by the state to the farmers. Therefore, to improve and enhance the rural human settlement environment, governments must make public policy and financial investments. The microlevel evaluation of farmers’ satisfaction is an important way to understand the current situation of the rural human settlement environment and guide its further construction [17]. However, the effect of farmers’ subjective satisfaction with the government-guided human settlement environment on their well-being has not yet received sufficient attention.

Nature reserves are key for maintaining people’s ecological well-being and are crucial for biodiversity conservation [18]. China’s nature reserves cover a total area of 1,471,700 square kilometers, accounting for 14.86% of the country’s land area. By 2035, nature reserves are expected to cover 18% of China’s land. Liaoning Province, located in northeastern China, has 104 nature reserves; these provide essential ecological support for maintaining ecological security in the northeast region. Nature reserves are surrounded by complex social and ecological systems. The goal of establishing nature reserves has evolved from simply pursuing ecological benefits to rebuilding the human–nature relationship and providing benefits to local farmers while protecting ecological resources [19,20]. The conservation of China’s natural ecosystems has entered a new phase of “health, stability, and efficiency”. To achieve rural revitalization and the construction of nature reserve systems, it is necessary to protect the welfare of farmers living around nature reserves and actively explore ways to improve their well-being. Moreover, this initiative is also crucial for shifting from “natural resources” to “economic benefits.” However, research on the subjective well-being of farmers around nature reserves is scarce.

Based on government-guided human settlement environment construction, this study classifies human settlement environments into three categories: natural ecological environment, hardware facility environment, and daily governance environment. Further, this study has been conducted to assess the influence of the rural human settlement environment on subjective happiness, to examine the construction achievements of China in rural human settlement environments, and to identify the impacting mechanism for such well-being. The highlights of main significance of this study are summarized as follows. First, as this study adopts a novel research perspective, it broadens the scope of research on this topic and supplements previous related studies. Second, this study has a precise research objective. The level of well-being of farmers living in communities around nature reserves has been shown as a key factor affecting the achievement of biodiversity conservation goals [21]. This study enriches the research population through a survey of 1002 rural households within and outside nature reserves. Third, this study has strong policy implications. It analyzes the effect of the rural human settlement environment around nature reserves on farmers’ well-being in the context of nature reserve system construction and rural revitalization. Identifying breakthroughs to enhance farmers’ subjective well-being and evaluating the construction achievements of rural human settlement environments will provide realistic policy recommendations for the construction initiative of rural human settlement environments.

## 2. Theoretical Background and Research Hypotheses

According to Maslow’s hierarchy of needs theory, once people’s physiological needs are satisfied, they will seek the satisfaction of material and noneconomic needs beyond a basic level. Farmers living in the vicinity of nature reserves hope that the high-quality ecological environments they guard will bring them ecological well-being and that they will enjoy equal access to urban and rural infrastructures and see improvements to the dirty, disorderly, and badly kept rural living environment [22]. However, in the context of an unbalanced urban–rural development, farmers are often at a disadvantage in the sharing of the dividends of economic development and social progress, which is characterized by a relatively backward living environment, a damaged natural ecological environment, weak hardware facilities, and poor daily governance. Therefore, satisfaction with the natural ecological environment, hardware facilities, and daily governance is crucial to gain satisfaction with the human settlement environment. Based on the “domain theory” posited by American social psychologist Kurt Lewin, French and Kahn proposed the “supply-value matching” theory, which argues that higher-level matching between individual preferences and the environment indicates a stronger sense of well-being [23,24]. Additionally, a better match between farmers’ settlement environment demand and actual supply will contribute to a higher level of well-being, thus evading the “supply-demand deviation effect”. Thus, Hypothesis 1 was proposed.

**Hypothesis** **1** **(H1).**
*Higher levels of satisfaction with the human settlement environment contribute to a stronger well-being among farmers living around nature reserves.*


Natural ecosystems are the foundation of a human settlement such that the topography, climate, hydrology, and vegetation are the natural substrates that make up a region’s human settlement environment, and they are related to the physical, mental, and ecological well-being of rural residents [25]. Nature reserves are an effective means of maintaining and enhancing ecosystem services [26]. Adequate ecosystems enable farmers living around nature reserves to enjoy regulating services, such as air and water quality improvement, and supporting services, such as soil and vegetation restoration. The communal nature of property rights and the nonexclusive nature of public benefits in ecosystems suggest that improving the natural ecosystem is an important mechanism to directly enhance the farmers’ well-being [27]. Therefore, Hypothesis 2 was proposed.

**Hypothesis** **2** **(H2).**
*Higher levels of satisfaction with the natural ecosystem contribute to a stronger well-being among farmers living around nature reserves.*


Furthermore, rural hardware facilities are an important factor influencing subjective well-being and are central to a human settlement environment. In recent years, the central government has shown increased support for the construction of rural human settlement environments, shifting focus from highly competitive private consumption to public expenditure that can be shared by all, resulting in improved rural hardware facilities. These measures have countered the deterioration to well-being caused by the bandwagon effect, alleviated farmers’ “relative deprivation” caused by the inequitable distribution of income, reduced precautionary savings and increased current consumption, and exerted a positive effect on the farmers’ well-being [28,29]. Improvements in facilities such as health services, public education resources, road transport, recreational centers, and green infrastructure contribute to the residents’ subjective well-being [30,31,32]. Based on the above analysis, Hypothesis 3 was proposed.

**Hypothesis** **3** **(H3).**
*Higher levels of satisfaction with the hardware facility environment contribute to a stronger well-being among farmers living around nature reserves.*


In addition, daily governance, which is a dynamic process characterizing good governance action at the grassroots level, is a crucial aspect of rural human settlement environmental governance. “Living environmentalism” suggests that environmental decline can be reversed by intervening in the behavior of people who are at the “base of environmental governance”. When the intensive and recurring practices of daily life are assigned connotations in environmental governance, the reconfiguration of practices can transform regular daily life into an environmentally friendly life [33]. The effect of rural governance quality on farmers’ well-being is more significant than that of individual characteristic elements [34]. Therefore, the organic combination of a human settlement environment and daily governance in rural areas is essential to obtain tangible governance effects. Additionally, adequate quality of daily governance can provide local farmers with an environment suited to their needs, thus improving their well-being. Thus, Hypothesis 4 was proposed.

**Hypothesis** **4** **(H4).**
*Higher levels of satisfaction with the daily governance environment contribute to a stronger well-being for farmers living around nature reserves.*


## 3. Materials and Methods

### 3.1. Data Sources

This study conducted a field survey of farmers living around nature reserves in Liaoning Province from July to August 2021. Nature reserves in Liaoning Province cover an area of 2.44 million hectares, accounting for approximately 11.1% of the province’s land area. Among the nature reserves, 18 are national-level and 27 are provincial-level reserves. Farmers around nature reserves include those within and outside the reserves. Farmers within nature reserves are those living in the core areas, buffer zones, and pilot areas of nature reserves, whereas most farmers outside the reserves are villagers within 20 km of the periphery of nature reserves, whose livelihoods are not dependent on nature reserves [20]. In this study, three national-level and three provincial-level nature reserves were selected using stratified random sampling, as shown in Figure 1. The national-level nature reserves selected were Liaoning Laotu Dingzi National Nature Reserve, Fuxin Haitanshan National Nature Reserve, and Dandong Baishi Lizi National Nature Reserve. The provincial-level reserves included Fushun Sankuaishi Nature Reserve, Benxi Heshang Maozi Nature Reserve, and Fushun Monkey Rock National Forest Park. Moreover, 17 and 27 villages within and outside the nature reserves, respectively, were randomly selected. Next, 25 villagers were randomly selected from each of the 44 selected villages. We ensured, as far as possible, that the respondents were the head of the household or their spouse and were familiar with the household’s income and expenditure. In addition, all farmers within the Monkey Rock National Forest Park, Laotu Dingzi, and Baishi Lizi National Nature Reserves had completed their ecological migration. The questionnaires were then completed through face-to-face interviews with the selected farmers. In total, 1002 questionnaires were returned. A total of 951 valid questionnaires were obtained after screening out invalid questionnaires and those with missing data, with an effective response rate of 94.91%. Among them, 402 and 549 questionnaires were completed by farmers within and outside the nature reserves, respectively (Table 1).

### 3.2. Variable Selection

The explained variable in this study was the well-being of farmers living around nature reserves. In the survey on individual well-being, the respondents were asked the following question: “Overall, do you feel that you are happy?” The responses were recorded on a 5-point Likert scale ranging from 1 to 5 with the following response options: very unhappy (1), unhappy (2), average (3), happy (4), and very happy (5). Higher scores indicated a greater well-being. Figure 2 shows the average happiness of 44 villages inside and outside the nature reserves. In the whole research area, the average happiness index of farmers was 4.00, the average happiness outside the nature reserves was 3.96, and the average happiness inside the nature reserves was 4.06. There were 16 villages with an average happiness greater than 4.00, accounting for 59.3% of all the villages outside the nature reserve. Moreover, there were 12 villages with an average happiness greater than 4.00, accounting for 70.6% of the total villages in the reserve. The five villages with the lowest average happiness were all outside the reserves.

The core explanatory variables included the natural ecological, hardware facility, and daily governance environments (Table 2). We drew on existing studies and the *Three-Year Action Plan for the Improvement of Rural*
*human settlement*
*and the Rural Construction Initiative* that strengthens agricultural and rural infrastructure and the construction of human settlement environments as the basis of our research. Additionally, an evaluation system for the human settlement environment focusing on the governance level of the government was constructed, following the principles of scientificity, systematicity, comprehensiveness, comparability, and operability, and taking into account the basic requirements for the ecological protection of nature reserves.

This study drew on the studies of Zhao and Zhang (2006), Li (2015), and Hao et al. (2020); four indicators, namely, air quality, wildlife population, water conservation, and soil and vegetation restoration were selected to characterize the natural ecological environment [25,35,36]. The governance of the natural and ecological environment in rural areas is one of the general requirements for rural revitalization. Air quality is a significant indicator of a good natural and ecological environment and possesses important practical significance for residents’ happiness. Studies have identified an intricate link between air quality and residents’ well-being in the United States and Europe. Moreover, air pollution adversely impacts residents’ happiness [37,38]. Water conservation, soil, and vegetation are sine qua non factors in the establishment of a living environment, and they also reflect the service value of ecosystems in nature reserves. The MA report by the United Nations states that there is a strong relationship between ecosystem services and human well-being [39]. The construction of nature reserves has boosted the ecological environment, and the number of wild animals has increased year by year. Therefore, conflicts between farmers around the nature reserves and the animals have been exacerbated, exerting a certain impact on farmers’ happiness [40].

**Table 2 ijerph-19-06447-t002:** Evaluation system for rural human settlement environments.

		Total Samples		Samples Inside the Reserves		Samples Outside the Reserves	
Primary Indicators	Secondary Indicators	Mean	Standard Deviation	Mean	Standard Deviation	Mean	Standard Deviation
Natural ecological environment	Air quality	4.327	0.752	4.237	0.750	4.393	0.748
	Wildlife population	3.891	1.057	3.765	1.103	3.983	1.013
	Water conservation	3.706	1.233	3.495	1.268	3.859	1.185
	Soil and vegetation restoration	3.887	0.984	3.770	0.971	3.972	0.985
Hardware facility environment	Medical service facilities	3.534	1.171	3.507	1.203	3.553	1.147
	Cultural, sports, and recreational facilities	3.606	1.151	3.637	1.153	3.584	1.150
	Basic living facilities	3.906	1.04	3.895	1.073	3.914	1.017
	Environmental beautification facilities	4.02	1.024	4.062	0.995	3.989	1.044
Daily governance environment	Waste disposal	4.057	1.042	3.920	1.114	4.158	0.975
	Community security	4.445	0.646	4.440	0.622	4.449	0.664
	Drinking water quality	3.812	1.227	3.570	1.274	3.989	1.162
	Convenience of living	4.146	0.979	4.190	0.906	4.114	1.028

Note: Basic living facilities include water, electricity, roads, gas, radio and television, communications, logistics, and other facilities; environmental beautification facilities include brightening and landscaping and other facilities; the degree of convenience of living refers to the accessibility of services such as the living circle, service circle, and business circle [41].

Following the studies of Zhang and Xu (2020) and Wei et al. (2021), hardware facilities were defined as medical service facilities; basic living facilities; cultural, sports, and recreational facilities; and environmental beautification facilities [31,41]. Numerous studies mention that public medical services and basic living facilities can improve residents’ well-being [28]. Neil and Huw (2008) concluded that diversified public activity venues and sophisticated recreational facilities could improve residents’ life quality and happiness [42]. The improvement of a human settlement environment is a long-term process aimed at beautifying the countryside. Environmental beautification facilities were incorporated into the hardware facility indicator system, as linking these two can facilitate the creation of happy living spaces with greening, beautifying, and brightening elements. Green infrastructure is an integral part of landscaping facilities. It fulfills the functions of raising residents’ health levels and well-being levels by soothing residents’ mental pressure, enhancing rainwater management capabilities, and improving the built environment. Therefore, it leaves a profound impact on residents’ well-being [43,44,45]. Furthermore, education facilities at the village level were not included in the indicator system because increasing numbers of children are migrating from villages to urban areas, leading to an expansion of boarding schools in townships, while simultaneously increasing the number of vacant schools in villages.

Based on the findings of Ballas and Tranmer (2012), Huang (2020), Wei et al. (2021), and Xiao et al. (2021), waste disposal, drinking water quality, security, and convenience of living were selected as daily governance indicators in this study [41,46,47,48]. In the context of rural revitalization, waste and sewage treatment facilities are the fundamental hardware for improving the quality of the human environment; thus, they should be given adequate consideration [49]. As most villages are equipped with fixed waste drop-off points and use collection and transfer methods for waste disposal, farmers have a clear perception of the daily waste removal and disposal chain. Further, drinking water quality was chosen as a secondary indicator in this study, as rural wastewater treatment facilities are located in townships and are responsible for ensuring the safety of drinking water for rural residents [50]. In addition, security in rural communities is a major component of rural community building; good security conditions enhance individuals’ level of subjective well-being [51]. Additionally, building convenient commercial network circles and improving service circles such as postal and express delivery can contribute to a more comfortable life and is also key in the construction of human settlement environments [41]. All four variables were measured on a 5-point Likert scale, ranging from 1 (very poor) to 5 (very good).

Multiple control variables were included reflecting both individual and household traits. Individual traits included gender age, health status, educational attainment, and interpersonal relationships. Household traits included household income status, value of the household house, and area of the household forest land (Table 3).

### 3.3. Model Setting

This study established the following master model to evaluate the effect of satisfaction with the human settlement environment on farmers’ well-being:(1)$$h_{i}=\alpha +\lambda \times S_{i}^{k}+\mu \times x_{i}+\varepsilon_{i}$$
where *h_i_* is individual well-being, λ is the regression coefficient of *S_i_^k^*, *S_i_^k^* is the evaluation of the *k*th type of human settlement environment (*k* = 1, 2, 3, for the natural ecological environment, hardware facility environment, and daily governance environment, respectively), *x_i_* denotes the control variables at the level of personal traits and household traits, and μ is the estimated coefficient of *x_i_*. The questionnaire employed in this study used ordinal numbers as a measure of the farmers’ satisfaction with natural ecology environment, hardware facility, and daily governance and equal weights to calculate the indices. Additionally, well-being was assessed using ordinal numbers and an econometric analysis was conducted using the ordered probit regression model and OLS model. As these two methods have identical measurement results in terms of significance level and sign direction of the explanatory variables, the OLS estimation was eventually included in this study due to its convenience of analysis.

To investigate whether there were any differences in the mechanism of the effect of rural human settlement environment on farmers’ well-being within and outside the nature reserve, regression analyses were conducted using secondary indicators of human settlement environment separately and the following model was obtained:(2)$$h_{i}=\beta +\delta \times S_{i}^{j}+\varphi \times x_{i}+\varepsilon_{j}$$
where *σ* is the regression coefficient of *S_i_^j^*, *S_i_^j^* is the evaluation of the *j*th secondary indicator, *x_i_* denotes the control variable at the individual and household trait levels, and φ is the estimated coefficient of *x_i_*.

## 4. Results

### 4.1. Human Settlement Environment and Well-Being

This section presents the empirical evaluation of the four hypotheses proposed in this study. Table 4 presents the regression results of the master model (1) where columns (1)–(3) show the results of regression for the natural ecological environment, hardware facility environment, and daily governance environment, respectively. The explanatory variables for column (4) include the regression results for the three types of environmental satisfaction. The regression results for the core variables of columns (1)–(3) suggested that each of the three types of environment—natural ecological environment, hardware facility environment, and daily governance environment—had a significant positive effect on farmers’ well-being at the 1% level of significance, validating H2, H3, and H4, respectively. Hence, higher satisfaction levels with the human settlement environment contributed to a higher well-being, validating H1.

The marginal effects of satisfaction with the natural ecological environment and daily governance environment on well-being are roughly equal (λ = 0.109 and 0.106, respectively). Satisfaction with hardware facilities shows the greatest marginal effect on well-being (λ = 0.126). A comparison of the regression coefficients of satisfaction with each type of human settlement environment in column (4) with those in the other three equations suggests that the regression coefficients of all three types of human settlement environment decreased. Therefore, there are positive correlations among the three types of human settlement environment. The coefficient of daily governance environment in column (4) was not significant, indicating that daily governance environment fails to improve farmers’ well-being.

For the control variables, the regression results show that educational attainment is not significant in relation to the area of the household forest land and the value of the family house, whereas all other variables are significant to varying degrees. The regression coefficient for gender indicates a significant effect of sex on farmers’ well-being; women’s well-being is generally higher than that of men. The regression coefficient for age indicates that there is a significant positive association between age and well-being. Additionally, health status has a significant positive effect on well-being. Furthermore, column (4) shows that the relationship with the village committee has a significant positive effect on well-being; it has the largest marginal effect and twice the effect on well-being compared to that of the natural ecological environment. The total annual household income has a significant positive effect on farmers’ well-being. In addition, the effect of the area of the household forest land is significant at the 10% level with a negative sign.

### 4.2. Robustness Tests

The measurement and sample replacement methods were employed in this study to test the robustness of the obtained empirical results. Columns 1–4 in Table 5 present the regression results using the measurement replacement method; the effects of the natural ecological environment, hardware facility environment, and daily governance environment are all significant at the 1% level (with positive signs). The significance levels and signs of the explanatory variables in column 4 are consistent with the aforementioned master model, suggesting that satisfaction with the human settlement environment significantly improves farmers’ well-being and confirming the robustness of the master model. However, it must be noted that the present results may have been influenced by the precise poverty alleviation policy that was implemented in China during the same period. This policy has simultaneously achieved poverty incidence reduction and poverty alleviation and led to significant improvements in both the income level and quality of life of rural residents, indirectly increasing farmers’ well-being [51]. To control for the interference of this policy, the samples of rural households in Fuxin, the city where the pilot experiment of poverty alleviation was conducted, were removed. Thereafter, the regression analysis was performed again. The specific results are shown in columns 5–8 in Table 5. The significance and signs of the regression coefficients of each variable are consistent with the previous results, confirming the robustness of the model.

### 4.3. Heterogeneity Analysis

To further explore the differences in the effect of farmers’ human settlement environment satisfaction on their well-being, this study adopted an ordered probit model. Individual and household trait-related control variables were added to each column. The specific regression results are shown in Table 6. Satisfaction with the natural ecological environment, hardware facility environment, and daily governance environment each has a significant positive effect on farmers’ well-being. For farmers within nature reserves, the results in columns 1–3 indicate that satisfaction conditions with the natural ecological environment and hardware facility environment have the largest and smallest regression coefficients, respectively (σ = 0.244 and 0.157, respectively), indicating that satisfaction with the natural ecological environment has the largest marginal effect on the well-being of farmers within nature reserves. For farmers outside nature reserves, the results in columns 5–7 show that satisfaction conditions with the hardware facility environment have the largest marginal effect on their well-being (σ = 0.224). Additionally, the natural ecological environment, and daily governance environment have similar marginal effects on well-being. The results in columns 4 and 8 of Table 6 show that satisfaction with the natural ecological environment has the most significant effect on the well-being of farmers within nature reserves (σ = 0.191). In contrast, satisfaction with the hardware facility environment has the most significant positive effect on the well-being of farmers outside nature reserves (σ = 0.192).

Ed Diener et al. (2009) revealed that increasing public expenditure can enhance residents’ well-being when the government has a clear understanding of their preferences [52]. To investigate the heterogeneity of the effect of farmers’ satisfaction preferences on their well-being, this study incorporated human settlement environment secondary indicators in turn into Model 2 for empirical testing. Furthermore, individual and household trait-related control variables were added to each column. The specific regression results are shown in Table 7, Table 8 and Table 9.

#### 4.3.1. Heterogeneity Analysis of the Effect of Natural Ecological Environment on Well-Being

The regression results of the effect of the natural ecological environment on farmers’ well-being are presented in Table 7. The “total sample” column in Table 7 shows that air quality, wildlife population, and soil and vegetation restoration have positive and significant effects on farmers’ well-being. However, water conservation does not have a significant effect on well-being. For farmers within the nature reserves, all indicators significantly increase their well-being, except wildlife population, which does not have a significant effect on their well-being and the sign of the regression coefficient is negative. For farmers outside the nature reserves, the most significant indicator of well-being is wildlife population, whereas all other indicators are nonsignificant.

#### 4.3.2. Heterogeneity Analysis of the Effect of Hardware Facility Environment on Well-Being

The regression results of the effect of the hardware facility environment on farmers’ well-being are shown in Table 8. The “total sample” column in Table 8 suggests that satisfaction with all types of hardware facilities significantly and positively influences farmers’ well-being, indicating that improvements in the hardware facility environment are a major factor contributing to farmers’ satisfaction with the human settlement environment. Satisfaction with the construction of cultural, sports, and recreational facilities and environmental beautification services significantly and positively influences the well-being of farmers within the nature reserves. However, medical service facilities and basic living facilities do not have a significant effect on the well-being of farmers within the nature reserves. In contrast, the various types of hardware facilities have a significant positive effect on the well-being of farmers outside the nature reserves.

#### 4.3.3. Heterogeneity Analysis of the Effect of Daily Governance Environment on Well-Being

Table 9 presents the results of regression showing the effect of daily governance environment on well-being. The regression results of the total sample suggest that neither waste disposal nor drinking water quality has a significant effect on well-being. However, community security and convenience of living have significant positive effects on farmers’ well-being. Furthermore, waste disposal and drinking water quality did not pass the empirical test.

## 5. Discussion

The findings of the present study help to reveal the mechanisms that influence farmers’ well-being. Actively exploring ways to enhance the well-being of farmers around nature reserves is important for achieving rural revitalization and the construction of nature reserve systems in China. Studies on well-being conducted in China and abroad have focused mainly on the influence of individual or household traits, such as self-perception, interpersonal relationships, and family decision-making, and public policies, such as social hardware facilities, ecological environment, and policy governance [53,54,55,56]. However, the effect of the rural human settlement environment on farmers’ well-being has been ignored. This study constructed an evaluation model of farmers’ human settlement environment satisfaction from a micro perspective and analyzed the effect of human settlement environment satisfaction on the well-being of farmers within and outside nature reserves. Additionally, a heterogeneity analysis was conducted. Overall, the findings of this study broaden the scope of well-being-related research and enrich the field of research. Moreover, our findings may facilitate the improvement of the efficiency of China’s rural human settlement environment construction, enhance farmers’ well-being, and ultimately help achieve the goal of China’s rural revitalization strategy.

Natural ecological environment is a stabilizer of farmers’ well-being. In particular, attention should be paid to the regression results of two variables, area of household forest land and wildlife population. In general, a larger household forest land area indicates richer forestry resources at the household’s disposal and consequently, greater welfare for the household [57]. However, in this study, the area of household forest land had a negative effect on farmers’ well-being. This may be because owing to a series of national measures to strengthen rural ecological construction, it has become increasingly difficult for farmers outside nature reserves to apply for logging quotas; thus, a larger household forest land area indicates a greater loss of opportunity cost. According to the Regulations on the Management of Nature Reserves, farmers within nature reserves were unable to reap the economic benefits of harvesting forest trees. Moreover, some farmers had already contracted a large amount of commercial forest through forest land transfer and other means before the forest land was designated as a nature reserve. Therefore, they strongly demanded that their reasonable rights and interests in forest land management be safeguarded. Thus, the regression results for household forest land area in this study are nonsignificant and have a negative sign. Therefore, adequate handling of the contradiction between the preservation of nature reserves and local farmers’ demands is one of the ways to enhance farmers’ well-being. In the heterogeneity analysis, satisfaction with the natural ecological environment had a more significant effect on enhancing the well-being of farmers inside nature reserves than of those outside nature reserves. In practice, these farmers are participants in the construction of the nature reserve system. Satisfaction with air quality, water conservation, and soil and vegetation restoration endorses ecological conservation and significantly enhances farmers’ well-being. The wildlife population had a negative effect on farmers’ well-being, which is in line with the findings of Hou and Wen (2012) and Bluwstein et al. (2016) [58,59]. The construction of nature reserves has forced farmers in protected areas to change their traditional livelihood modes and has increased livestock farming costs, ultimately constraining land development rights. In addition, some farmers have not yet received a reasonable ecological compensation. Therefore, a good natural ecological environment may not only affect farmers’ well-being positively but may conversely have a negative effect on their well-being by hindering the economic development of the local community or increasing their production and operation costs [60].

Moreover, higher ecological control results in the growth of wildlife populations in nature reserves, which in turn results in wildlife perpetration on the crops and livestock of surrounding villages, leading to increasing human–wildlife conflict. This may explain why in the present study, the regression coefficient for the effect of the wildlife population indicator on well-being was negative. Human–animal conflicts in nature reserves are common [61]. Nowadays, an excessive number of wild animals in some parts of China have resulted in the saturation of their original management capacity in human settlement environments. Therefore, the scope of their activities is continuing to expand, and instances of damage perpetrated by them will occur sporadically. According to incomplete statistics by the provinces in China, 3,811,300 incidents perpetrated by wild animals occurred from 2017 to 2020, causing direct economic losses of over 15.38 billion yuan [38]. As of July 2021, a total of nine provinces (cities and regions) had issued compensation measures for damage incurred by local terrestrial wildlife. Due to varying natural and economic conditions in different places, the compensation standards fluctuate: governments of some underdeveloped areas have only limited financial resources and lack special compensation funds for wildlife-induced damage; and some governments only provide compensation for the damage caused by the protected animals, excluding the damage to crops caused by other animals [58]. The Chinese government has made progress in relieving the human–wildlife conflicts in nature reserves; however, there are still numerous issues in terms of the wholesomeness of relevant laws and regulations, the compensation standards, and the insurance damage assessments. For farmers outside nature reserves, the increasing number of wild animals indicates a good ecological environment for farmers outside nature reserves. Given that wild animals mainly inhabit remote nature reserves, there are few incidents of wild animals spoiling livestock or destroying crops. Therefore, wild animals have a significant positive impact on the well-being of farmers outside nature reserves.

Improving farmers’ satisfaction with hardware facilities is important for enhancing their well-being. Consistent with Ma’s (2018) findings, both cultural, sports, and recreational facilities and environmental beautification facilities significantly and positively affected the well-being of farmers both inside and outside the nature reserves [62]. Improvements in cultural, sports, and recreational facilities uphold farmers’ right to development and guarantee their cultural freedom. Additionally, environmental beautification facilities satisfy farmers’ need to enrich their inner world and promote their personal spiritual development. Therefore, higher levels of satisfaction with cultural, sports, and recreational facilities and environmental beautification facilities indicate a greater sense of well-being among farmers both within and outside nature reserves. Currently, Chinese farmers’ demand for hardware facility environment has changed from basic facilities related to livelihood to developmental facilities that enhance their quality of life.

Satisfaction with the daily governance environment contributes positively to farmers’ well-being. However, the regression results for the effect of two secondary indicators—waste disposal and drinking water safety—on well-being are worth considering. Environmental governance has an innovative effect. According to the “cost-compliance theory,” in the short term, innovation may indirectly reduce residents’ well-being by increasing the cost of governance and inhibiting local economic growth due to immature technological innovation [63,64]. The daily management of a rural human settlement environment is currently restricted in terms of lack of standardized technologies, large funding gaps, and inadequate institutional mechanisms [65], leading to a “governance dilemma” at the rural grassroots level. Therefore, its role in farmers’ well-being needs to be further explored. The management of rural domestic waste and sewage is a core component of human settlement environment improvement. However, currently, there is a disconnect between knowledge of waste disposal and farmers’ values and practices. Additionally, waste disposal is a high-cost project, which leads to “exhaustion” in rural governance and may even have a negative effect on well-being [66]. Another secondary indicator, convenience of living, has a positive and significant effect on farmers’ well-being, with the largest marginal effect. Convenience of living is key for improving farmers’ human settlement environment and is the focus of future human settlement environment efforts.

To further advance rural revitalization and enhance the well-being of different regional groups through the enjoyment of an ever-improving living environment, this study provides the following five policy insights: (1) Formulating environmental protection policies and promoting public liability insurance for wildlife accidents in nature reserves to improve the compensation mechanism for damages caused by wildlife; (2) Improving the hardware environment by focusing on improving cultural, sports, and recreational facilities and environmental beautification facilities to ensure farmers’ well-being; (3) Optimizing the daily environmental governance mechanism at the grassroots level, and popularizing relevant knowledge to improve the governance efficiency by grassroots organizations; (4) Establishing priorities for human settlement environment and improving the efficiency of decision-making to effectively allocate collective economic resources to promote rural revitalization and improve farmers’ well-being, as human settlement environment improvement is a challenging and long-term process; (5) Identifying the restrictions of human settlement environment construction in the regions and the demands of local farmers to determine the priorities. In addition, focusing on the “precision” of human settlement environments can generate an adequate response to the expectations of farmers in different regions.

This study has some limitations. First, the mechanism of the heterogeneous influence of satisfaction with the human settlement environment on farmers’ well-being lacks in-depth exploration and needs further research. Second, household toilet renovation is an important task in the construction of a human settlement environment. However, in Liaoning Province, which is located in northeastern China, the implementation of renovation projects has not been effective due to winter weather conditions and technical constraints. Therefore, this important variable has not been included as a control variable in this study.

## 6. Conclusions

The study empirically analyzed the effect of human habitat and human settlement environment construction on farmers’ well-being in three aspects, natural ecological environment, hardware facility environment, and daily governance environment, using data from 1002 farmers living within and outside six nature reserves in Liaoning Province, China, in 2021. Our findings suggested that farmers’ satisfaction with the human settlement environment could significantly increase their well-being. Satisfaction with the natural ecological environment, hardware facility environment, and daily governance environment each significantly and robustly contributed to well-being. The hardware facility environment had the largest marginal effect on farmers’ well-being. Robustness testing showed that the main findings of this study remained unchanged. A heterogeneity analysis showed that the natural ecological environment played a crucial role in enhancing the well-being of farmers inside nature reserves, while the hardware facility environment had the strongest effect on the well-being of farmers outside nature reserves. Overall, this study helps to indicate the direction and improve the efficiency of human settlement environment construction and enhance its positive effect on well-being. Moreover, the findings of this study have important practical implications for improving farmers’ well-being and achieving the goals of rural revitalization in China.

## Figures and Tables

**Figure 1 ijerph-19-06447-f001:**
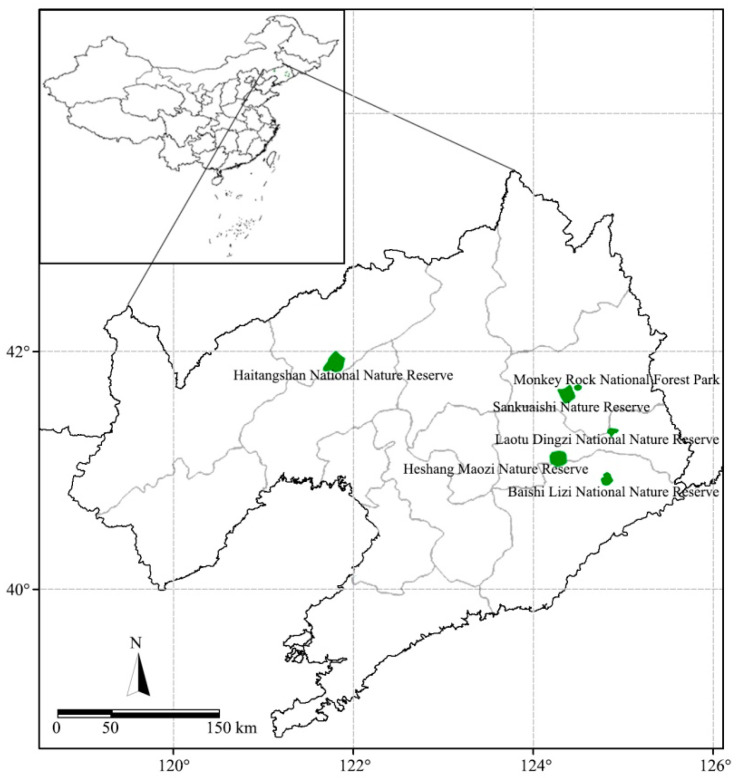
Six nature reserve locations in Liaoning Province of China.

**Figure 2 ijerph-19-06447-f002:**
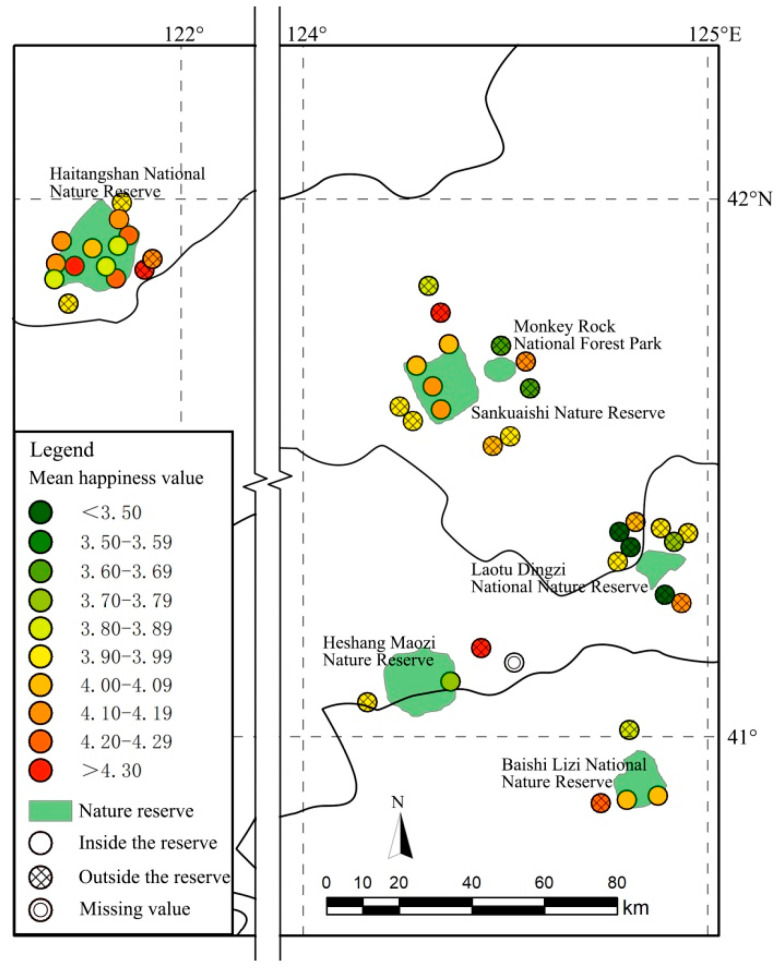
Average happiness of 44 villages inside and outside the nature reserves.

**Table 1 ijerph-19-06447-t001:** Sample statistics.

Level	Natural Reserve	Research Sites	Sample Size	Percentage (%)
National level	Haitanshan Nature Reserve	Inside the reserve	253	26.60
		Outside the reserve	59	6.20
	Laotu Dingzi Nature Reserve	Inside the reserve	0	0
		Outside the reserve	184	19.35
	Baishi Lizi Nature Reserve	Inside the reserve	0	0
		Outside the reserve	74	7.78
Provincial level	Sankuaishi Nature Reserve	Inside the reserve	98	10.30
		Outside the reserve	94	9.88
	Heshang Maozi Nature Reserve	Inside the reserve	51	5.36
		Outside the reserve	19	2.00
	Monkey Rock National Forest Park	Inside the reserve	0	0
		Outside the reserve	119	12.51

**Table 3 ijerph-19-06447-t003:** Names, definitions, means, and standard deviations of variables.

Variables	Definition	Mean	Standard Deviation
Explained variable			
Well-being	Very happy, 5; happy, 4; fair, 3; unhappy, 2; very unhappy, 1	4.004	0.7377
Explanatory variables			
Satisfaction with natural ecological environment	Mean value of satisfaction with air quality, wildlife, water conservation, and soil and vegetation restoration	3.953	0.700
Satisfaction with hardware facility environment	Mean value of satisfaction with medical services, cultural, sports and recreational, basic living, and environmental beautification facilities	3.766	0.828
Satisfaction with daily governance environment	Mean value of satisfaction with waste and sewage disposal, community security, drinking water quality, and convenience of living	4.115	0.655
Control variables			
Gender	Male, 1; female, 0	0.577	0.494
Age (years)		54.75	10.62
Educational attainment of the head of household	No schooling, 1; primary school, 2; junior high school, 3; senior high school/technical secondary school, 4; junior college, 5; undergraduate, 6; postgraduate (including Ph.D. and above), 7	2.896	0.775
Health status	Very good, 1; good, 2; fair, 3; bad, 4; incapable of work, 5	1.751	0.904
Relationship with the village committee	very poor, 1; poor, 2; fair, 3; good, 4; Very good, 5	4.097	0.812
Total annual household income	Logarithm of the total income of all household workers	10.908	0.990
Area of the household forest land (mu)	Actual area of forest land operated by the household including self-reserved hills	69.949	222.993
Value of the household house (ten thousand yuan)	Total market value of all houses in the household (2021)	14.769	31.557

**Table 4 ijerph-19-06447-t004:** Satisfaction with human settlement environment and farmers’ well-being.

		(1)	(2)	(3)	(4)
Explained variable	Well-being				
Core explanatory variables	Satisfaction with natural ecological environment	0.109 ***			0.071 **
	Satisfaction with hardware facility environment		0.126 ***		0.104 ***
	Satisfaction with daily governance environment			0.106 ***	0.020
Individual traits	Gender	−0.122 **	−0.096 **	−0.110 **	−0.110 **
	Age	0.006 ***	0.005 **	0.006 ***	0.006 ***
	Educational attainment	0.054 *	0.038	0.048	0.045
	Health status	−0.120 ***	−0.126 ***	−0.117 ***	−0.125 ***
	Relationship with the village committee	0.187 ***	0.166 ***	0.182 ***	0.156 ***
Household traits	Total annual household income	0.093 ***	0.093 ***	0.091 ***	0.091 ***
	Area of the household forest land	−0.000 *	−0.000	−0.000	−0.000 *
	Value of the household house	−0.000	−0.000	−0.001	−0.001

Note: ***, **, and * represent significance at the 1%, 5%, and 10% statistical levels, respectively.

**Table 5 ijerph-19-06447-t005:** Regression results of the robustness test.

	Measurement Replacement Method (Ordered Probit)				Sample Replacement Method (Excluding Samples of a Poverty Alleviation Reform Pilot Area)			
	1	2	3	4	5	6	7	8
Satisfaction with natural ecological environment	0.183 ***			0.124 **	0.251 ***			0.196 **
Satisfaction with hardware facility environment		0.199 ***		0.161 ***		0.193 ***		0.146 **
Satisfaction with daily governance environment			0.172 ***	0.033			0.188 **	0.044
Control variables	Control	Control	Control	Control	Control	Control	Control	Control

Note: *** and ** represent significance at the 1% and 5% statistical levels, respectively.

**Table 6 ijerph-19-06447-t006:** Effect of human settlement environment on the well-being of farmers inside and outside nature reserves.

	Farmers inside Nature Reserves				Farmers outside Nature Reserves			
	Well-Being1	Well-Being2	Well-Being3	Well-Being4	Well-Being5	Well-Being6	Well-Being7	Well-Being8
Satisfaction with natural ecological environment	0.244 ***			0.191 **	0.174 **			0.113
Satisfaction with hardware facility environment		0.157 **		0.090		0.224 ***		0.192 ***
Satisfaction with daily governance environment			0.189 **	0.055			0.179 **	0.039
Control variables	Control	Control	Control	Control	Control	Control	Control	Control

Note: *** and ** represent significance at the 1% and 5% statistical levels, respectively.

**Table 7 ijerph-19-06447-t007:** Natural ecological environment and farmers’ well-being.

	Total Sample	Inside Nature Reserves	Outside Nature Reserves
	Well-Being	Well-Being	Well-Being
Air quality	0.126 ***	0.314 ***	0.027
Wildlife population	0.112 ***	−0.021	0.238 ***
Water conservation	0.021	0.079 *	−0.018
Soil and vegetation restoration	0.118 ***	0.191 ***	0.071
Control variables	Control	Control	Control

Note: *** and * represent significance at the 1% and 10% statistical levels, respectively.

**Table 8 ijerph-19-06447-t008:** Hardware facility environment and farmers’ well-being.

	Total Sample	Inside Nature Reserves	Outside Nature Reserves
	Well-Being	Well-Being	Well-Being
Medical service facilities	0.100 ***	0.057	0.131 ***
Cultural, sports, and recreational facilities	0.106 ***	0.088 *	0.116 ***
Basic living facilities	0.113 ***	0.053	0.155 ***
Environmental beautification facilities	0.114 ***	0.152 ***	0.086 *
Control variables	Control	Control	Control

Note: *** and * represent significance at the 1% and 10% statistical levels, respectively.

**Table 9 ijerph-19-06447-t009:** Daily governance environment and farmers’ well-being.

	Total Sample	Inside Nature Reserves	Outside Nature Reserves
	Well-Being	Well-Being	Well-Being
Waste disposal	0.010	0.025	−0.035
Drinking water quality	0.029	0.044	0.028
Community security	0.156 ***	0.106	0.010 ***
Convenience of living	0.167 ***	0.184 ***	0.156 ***
Control variables	Control	Control	Control

Note: *** represent significance at the 1% statistical levels, respectively.

## Data Availability

Not applicable.

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
