# Peer review of "Effect of Rural Human Settlement Environment around Nature Reserves on Farmers’ Well-Being: A Field Survey Based on 1002 Farmer Households around Six Nature Reserves in China"

_ijerph, 2022, doi:10.3390/ijerph19116447_

Round 1

Reviewer 1 Report

Quantitative research methodology (in this case, a survey) clearly articulates certain indispensable elements that should be present in a manuscript (J.W. Creswell; J.D. Creswell, 2018, Research Design: Qualitative, Quantitative, and Mixed Methods Approaches). By this, I mean the purpose, research questions, and hypotheses of the study are straightforward and lucidly presented. The authors in the "Introduction" section correctly outline what the research is about and what will be done in the manuscript. But the clearly stated (in one sentence) purpose is missing. 

Another shortcoming concerns the research questions that should be presented immediately after the statement of purpose, following quantitative research methodology (J.W. Creswell; J.D. Creswell, 2018, Research Design: Qualitative, Quantitative, and Mixed Methods Approaches). Sample question: What factors from the natural ecological environment influence farmers' well-being? Referring to survey research methodology, we first pose research questions, then build a survey questionnaire based on that. After presenting the research questions, we formulated hypotheses - which the authors correctly showed.

Author Response

Response to reviewer #1

Reviewer #1: Quantitative research methodology (in this case, a survey) clearly articulates certain indispensable elements that should be present in a manuscript (J.W. Creswell; J.D. Creswell, 2018, Research Design: Qualitative, Quantitative, and Mixed Methods Approaches). By this, I mean the purpose, research questions, and hypotheses of the study are straightforward and lucidly presented. The authors in the "Introduction" section correctly outline what the research is about and what will be done in the manuscript. But the clearly stated (in one sentence) purpose is missing.

Response: Thank you so much for your well-judged comments. I have revised as suggested. In the introduction part, the research purpose is added. The red font indicates the modified part, and the specific modification details are as follows:

“Further, this study has been conducted to assess the influence of rural human settlement environment on the subjective happiness to examine the construction achievements of China in rural human settlement environment, and identify the impacting mechanism for such well-being.”

Another shortcoming concerns the research questions that should be presented immediately after the statement of purpose, following quantitative research methodology (J.W. Creswell; J.D. Creswell, 2018, Research Design: Qualitative, Quantitative, and Mixed Methods Approaches). Sample question: What factors from the natural ecological environment influence farmers' well-being? Referring to survey research methodology, we first pose research questions, then build a survey questionnaire based on that. After presenting the research questions, we formulated hypotheses - which the authors correctly showed.

The answers to the research questions should be included in the "Conclusions" section.

Response: I have revised as suggested.

1- In the “Conclusions”, research questions is added. The specific modification details are as follows:

“The study empirically analyzed the effect of habitat human settlement environment construction on farmers' well-being in three aspects-natural ecological environment, hardware facility environment, and daily governance environment, using data from 1002 farmers living within and outside six nature reserves in Liaoning Province, China, in 2021.”

2- In the “Materials and Methods”, “Data sources” is palced in the first part. Please refer to the manuscript for details.

3- In the “Variable selection”, research questions have been posed first, then formulated theoretical analysis. the specific modification details are as follows:

“This study drew on the studies of Zhao & Zhang (2006), Li (2015), and Hao et al. (2020), while taking into account the suitability of micro-level evaluations, the relative consistency of topography and climate in the sample study area, and the logical linkages among the particular cases of nature reserves [35-36,26]. Furthermore, four indicators, namely, air quality, wildlife population, water conservation, and soil and vegetation restoration were selected to characterize the natural ecological environment [35-36,25]. The governance of the natural and ecological environment in rural areas is one of the general requirements for rural revitalization. Air quality is a significant indicator of a good natural and ecological environment and possesses important practical significance for residents' happiness. Studies have identified an intricate link between air quality and residents' well-being in the United States and Europe. Moreover, air pollution adversely impacts residents' happiness [37-38]. Water conservation, soil, and vegetation are sine qua non factors in the establishment of a living environment, and they also reflect the service value of ecosystems in nature reserves. The MA report by the United Nations states that there is a strong relationship between ecosystem services and human well-being [39]. The construction of nature reserves has boosted the ecological environment, and the number of wild animals has increased year by year. Therefore, conflicts between farmers around the nature reserves and the animals have been exacerbated, exerting a certain impact on farmers' happiness [40].

Following the studies of Zhang & Xu (2020) and Wei et al. (2021), hardware facilities were defined as medical service facilities; basic living facilities; cultural, sports, and recreational facilities; and environmental beautification facilities[31,41]. Numerous studies mention that public medical services and basic living facilities can improve residents' well-being [28]. Neil and Huw (2008) concluded that diversified public activity venues and sophisticated recreational facilities could improve residents' life quality and happiness [42].Habitat environment i Improvement of human settlement environment is a long-term process aimed at beautifying the countryside. Environmental beautification facilities were incorporated into the hardware facility indicator system, as linking these two can facilitate the creation of happy living spaces with greening, beautifying, and brightening elements. Green infrastructure is an integral part of landscaping facilities. It fulfills the functions of raising residents' health levels and well-being levels by soothing residents' mental pressure, enhancing rainwater management capabilities, and improving the built environment. Therefore, it leaves a profound impact on residents' well-being [43-45].Furthermore, education facilities at the village level were not included in the indicator system because increasing numbers of children are migrating from villages to urban areas, leading to an expansion of boarding schools in townships, while simultaneously increasing the number of vacant schools in villages.

Based on the findings of Ballas & Tranmer (2012), Huang (2020), Wei et al.(2021), and Xiao et al. (2021), waste disposal, drinking water quality, security, and convenience of living were selected as daily governance indicators in this study [46,47,41,48]. … …

Thank you so much for all of your comments and suggestions!

Reviewer 2 Report

In general, I believe this is a fine contribution to the scientific literature in this area and I commend the authors for undertaking this interesting and important research project. The study appears to me to be rigorous and worthy of publication, but I also believe the text can be substantially improved by reframing, reorganizing, and revising, and I suggest some awkward/problematic/confusing wording in English can be addressed with the help of a native English speaking editor. I recommend it be accepted with major revisions to the text, and below offer some questions and suggestions for improvement:

1 – the phrase “habitat environment construction” features in the title and is the first phrase mentioned in the abstract, but unfortunately has a confusing implication in English. I understand this word choice may be specific to the Chinese government documents mentioned in the paper. However, the meaning in English is unclear and to avoid losing or confusion readers and potential readers, I suggest another formulation. “Habitat” in English is usually paired with a specific user of the habitat (e.g., wildlife habitat, bird habitat, crane habitat, etc). Using the word habitat without reference to a user raises the question of what/who is the intended user of the habitat (ie a particular species or group of species). Furthermore, in English “habitat” almost always refers to plants or animals as users, rather than people. Because the focus of this paper is, in fact, rural people, I suggest dropping the word “habitat” in this context as that may clear up potential confusion in English. “Environment construction” is also not satisfactory, however, and so I suggest another term that may be suitable.

“Green infrastructure” is an existing term in the literature can be defined as “an inter-connected network of open, green spaces that provide a range of ecosystem services” ; or “any vegetative infrastructure system which enhances the natural environment through direct or indirect means” ; or “a strategically planned network of natural and semi-natural areas with other environmental features designed and managed to deliver a wide range of ecosystem services' in both rural and urban settings.” This term is recognized and used by Europeans and may be less familiar to Americans but makes more intuitive sense than “habitat environment construction,” which I strongly suggest replacing in order to draw more readers as well as to place this research in context with similar studies in Europe/Americas/elsewhere. I also suggest updating the literature review to refer to literature on “green infrastructure” and other similar related studies in this field.

2 – In the Abstract and the Methods sections of the paper, the analytical methods/models are mentioned before the reader is introduced to the field methods (interviews); it is standard and in my view more logical that the type of data being used should be mentioned ahead of how it was analyzed. Thus I would change both Abstract and Methods sections to change this order and mention the interviews and details about them ahead of the analytical approaches.

3 – The Introduction mentions the "hedonic treadmill theory," which is briefly explained, and goes on to mention the "well-being stagnation" dilemma, which is not explained but which the reader can guess is more or less the same as the hedonic treadmill theory (?). Although these terms are interesting for the reader to consider, I would caution the authors to adequately explain the theories they are introducing and to be very conservative in this regard; by this I mean if they can introduce their own hypotheses with less jargon or reference to obscure (and potentially entirely unsupported by empirical data (?); the reader is left to wonder), it is best for clarity and brevity to do so. Additional theories and speculation are better treated in the Discussion rather than the Introduction in my view.

4 – The Methods section of the paper contains the sentence, “This study drew on the studies of Zhao & Zhang (2006), Li (2015), and Hao et al. (2020), while taking into account the suitability of micro-level evaluations, the relative consistency of topography and climate in the sample study area, and the logical linkages among the  particular  cases  of nature  reserves  [35-36,26].”

Most readers will not be familiar with the studies mentioned, so that the authors should devote a sentence to each to explain their important. The “suitability of micro-level evaluations” also does not make sense unless context is provided; thus please reword this to clarify. “Logical linkages among the particular cases of nature reserves” also is a phrase that requires explanation/context. Please revise this sentence accordingly.

5 – In the Results section, the authors begin with the sentence, “This section presents the empirical evaluation of the four hypotheses proposed in this study. Table 4 presents the regression results of the master model.”

I would suggest replacing this with the paragraph below Table 4, which starts, “The regression results for the core variables of equations 1–3 suggest that each of the three types of environment—natural ecological environment, hardware facility environment, and daily governance environment—has  a significant positive effect  on farmers' well-being …”

I suggest this because it is preferable in my view to begin the Results section with a brief statement of the most important results. It should go without saying that the Results section presents empirical evaluation of the hypotheses proposed. Table 4 may be referenced in paratheses in the first sentence.

6 – In the Discussion section, the authors begin with the statement, “Attaining well-being is the ultimate goal of all human activities.” While this may be the authors’ opinion, others might disagree, depending on their beliefs, and in any event this statement is extraneous to this study’s hypotheses and findings and may distract from them, so I suggest deleting it.

Later in the same study the authors continue, “The findings of the present study help to reveal the mechanisms that influence farmers’ well-being. Actively exploring ways to enhance the well-being of farmers around nature reserves is important for achieving rural revitalization and ecological civilization in China.” Such statements, slightly revised, would be a much stronger way to start your discussion. The term “ecological civilization” does not quite make sense in English as ecology pertains to nature whereas civilization pertains to human culture. I suggest using a difference phrase to clarify what you mean.

7 – In the Discussion the authors also state, “… higher ecological control results in the growth of wildlife populations in nature reserves, which in turn results in wildlife perpetration on the crops and livestock of surrounding villages, leading to increasing human–wildlife conflict. This may explain why in the present study, the regression coefficient for the effect of the wildlife population indicator on well-being was negative. For farmers outside nature reserves, a good ecological environment has led to an increase in wildlife populations. However, such farmers did not experience the distress of wild animals destroying their livestock and crops, which is why wildlife population had a significant positive effect on their well-being.” Is this difference in the implications of growing wildlife populations inside or outside nature reserves due to the abilities of farmers to stop wildlife from destroying their crops, hunt wildlife (and thus control their populations) or something else? A discussion of this difference would be of great interest to wildlife scientists and those concerned with human-wildlife conflict so I suggest adding a paragraph or two to focus on this.

Author Response

Response to reviewer 2#

Reviewer #2: In general, I believe this is a fine contribution to the scientific literature in this area and I commend the authors for undertaking this interesting and important research project. The study appears to me to be rigorous and worthy of publication, but I also believe the text can be substantially improved by reframing, reorganizing, and revising, and I suggest some awkward/problematic/confusing wording in English can be addressed with the help of a native English speaking editor. I recommend it be accepted with major revisions to the text, and below offer some questions and suggestions for improvement:

1 – the phrase “habitat environment construction” features in the title and is the first phrase mentioned in the abstract, but unfortunately has a confusing implication in English. I understand this word choice may be specific to the Chinese government documents mentioned in the paper. However, the meaning in English is unclear and to avoid losing or confusion readers and potential readers, I suggest another formulation. “Habitat” in English is usually paired with a specific user of the habitat (e.g., wildlife habitat, bird habitat, crane habitat, etc). Using the word habitat without reference to a user raises the question of what/who is the intended user of the habitat (ie a particular species or group of species). Furthermore, in English “habitat” almost always refers to plants or animals as users, rather than people. Because the focus of this paper is, in fact, rural people, I suggest dropping the word “habitat” in this context as that may clear up potential confusion in English. “Environment construction” is also not satisfactory, however, and so I suggest another term that may be suitable.

“Green infrastructure” is an existing term in the literature can be defined as “an inter-connected network of open, green spaces that provide a range of ecosystem services” ; or “any vegetative infrastructure system which enhances the natural environment through direct or indirect means” ; or “a strategically planned network of natural and semi-natural areas with other environmental features designed and managed to deliver a wide range of ecosystem services' in both rural and urban settings.” This term is recognized and used by Europeans and may be less familiar to Americans but makes more intuitive sense than “habitat environment construction,” which I strongly suggest replacing in order to draw more readers as well as to place this research in context with similar studies in Europe/Americas/elsewhere. I also suggest updating the literature review to refer to literature on “green infrastructure” and other similar related studies in this field.

Response: Thank you so much for your well-judged comments. I have revised as suggested.

1- ” village habitat environment” were replaced with ”rural human settlement environment” throughout the manuscript. Please refer to the manuscript for details.

2- “Green infrastructure” literature review is added in the “Theoretical background” and “Variable selection”. The references have been updated at the same time. The red font indicates the modified part, the specific modification details are as follows:

(1) Theoretical background: “Improvements in facilities such as health services, public education resources, road transport, and recreational centers and green infrastructure contribute to residents' subjective well-being [30-32].”

(2) Variable selection: “Green infrastructure is an integral part of landscaping facilities. It fulfills the functions of raising residents' health levels and well-being levels by soothing residents' mental pressure, enhancing rainwater management capabilities, and improving the built environment. Therefore, it leaves a profound impact on residents' well-being [43-45].”

(3) References:

  1. Ying, J.; Zhang, X.J.; Zhang, Y.Q.; Bilan, S. Green infrastructure: systematic literature review. Econ. Res--Ekonomska Istraživanja. 2021, 2,1893202.
  2. Marques, P.; Silva, A.S.; Quaresma, Y.; Manna, L.R.; Neto, N.M.; Mazzoni, R. Home gardens can be more important than other urban green infrastructure for mental well-being during COVID-19 pandemics. Urban For Urban Gree 2021, 64, 127268.
  3. Nastran, M.; Pintar, M.; Železnikar, Š.; Cvejic, R. Stakeholders’ Perceptions on the Role of Urban Green Infrastructure in Providing Ecosystem Services for Human Well-Being. Land 2022, 11, 299.
  4. Liu, O.Y.; Russo, A. Assessing the contribution of urban green spaces in green infrastructure strategy planning for urban ecosystem conditions and services. Sustain Cities Soc. 2021,68, 102772.

2 – In the Abstract and the Methods sections of the paper, the analytical methods/models are mentioned before the reader is introduced to the field methods (interviews); it is standard and in my view more logical that the type of data being used should be mentioned ahead of how it was analyzed. Thus I would change both Abstract and Methods sections to change this order and mention the interviews and details about them ahead of the analytical approaches.

Response: I have revised as suggested.

1- The field method is added in the Abstract and posed before the analytical methods. The specific modification details are as follows:

“By adopting the method of random stratified sampling, this study investigates 1,002 farmers inside and outside 6 nature reserves in Liaoning, China. OLS and Ordered Probit regression models are used to assess the impact on the well-being of the satisfaction of farmers with their settlement environment around nature reserves from three aspects: the natural ecological environment; the hardware facility environment; and the daily governance environment.”

2- In the Materials and Methods, the order is adjusted. “Data sources” is palced in the first part. Please refer to the manuscript for details.

3 – The Introduction mentions the "hedonic treadmill theory," which is briefly explained, and goes on to mention the "well-being stagnation" dilemma, which is not explained but which the reader can guess is more or less the same as the hedonic treadmill theory (?). Although these terms are interesting for the reader to consider, I would caution the authors to adequately explain the theories they are introducing and to be very conservative in this regard; by this I mean if they can introduce their own hypotheses with less jargon or reference to obscure (and potentially entirely unsupported by empirical data (?); the reader is left to wonder), it is best for clarity and brevity to do so. Additional theories and speculation are better treated in the Discussion rather than the Introduction in my view.

Response: I have revised as suggested. Sentences containing either "hedonic treadmill theory" or “well-being stagnation” have been omitted. The specific modification details are as follows: “According to the "hedonic treadmill theory," Farmers' well-being brought about by a growth in wealth decreases as their income level increases [5-6]. Additionally, income inequality between urban and rural areas erodes farmers' well-being brought about by economic growth [7]. As the overall well-being of Chinese farmers has not yet reached a high level [8].simply increasing their income fails to improve their well-being and meet their actual needs [9].

4 – The Methods section of the paper contains the sentence, “This study drew on the studies of Zhao & Zhang (2006), Li (2015), and Hao et al. (2020), while taking into account the suitability of micro-level evaluations, the relative consistency of topography and climate in the sample study area, and the logical linkages among the  particular  cases  of nature  reserves  [35-36,26].”

Most readers will not be familiar with the studies mentioned, so that the authors should devote a sentence to each to explain their important. The “suitability of micro-level evaluations” also does not make sense unless context is provided; thus please reword this to clarify. “Logical linkages among the particular cases of nature reserves” also is a phrase that requires explanation/context. Please revise this sentence accordingly.

Response: I have revised as suggested. The new The paragraph is almost rewritten again. The specific modification details are as follows:

“This study drew on the studies of Zhao & Zhang (2006), Li (2015), and Hao et al. (2020), while taking into account the suitability of micro-level evaluations, the relative consistency of topography and climate in the sample study area, and the logical linkages among the particular cases of nature reserves [35-36,26]. Furthermore, four indicators, namely, air quality, wildlife population, water conservation, and soil and vegetation restoration were selected to characterize the natural ecological environment [35-36,25]. The governance of the natural and ecological environment in rural areas is one of the general requirements for rural revitalization. Air quality is a significant indicator of a good natural and ecological environment and possesses important practical significance for residents' happiness. Studies have identified an intricate link between air quality and residents' well-being in the United States and Europe. Moreover, air pollution adversely impacts residents' happiness [37-38]. Water conservation, soil, and vegetation are sine qua non factors in the establishment of a living environment, and they also reflect the service value of ecosystems in nature reserves. The MA report by the United Nations states that there is a strong relationship between ecosystem services and human well-being [39]. The construction of nature reserves has boosted the ecological environment, and the number of wild animals has increased year by year. Therefore, conflicts between farmers around the nature reserves and the animals have been exacerbated, exerting a certain impact on farmers' happiness [40].”

5 – In the Results section, the authors begin with the sentence, “This section presents the empirical evaluation of the four hypotheses proposed in this study. Table 4 presents the regression results of the master model.”

I would suggest replacing this with the paragraph below Table 4, which starts, “The regression results for the core variables of equations 1–3 suggest that each of the three types of environment—natural ecological environment, hardware facility environment, and daily governance environment—has  a significant positive effect  on farmers' well-being …”

I suggest this because it is preferable in my view to begin the Results section with a brief statement of the most important results. It should go without saying that the Results section presents empirical evaluation of the hypotheses proposed. Table 4 may be referenced in paratheses in the first sentence.

Response: I have revised as suggested. “The regression results for the core variables of equations 1–3 suggest…validating H1” is placed at the beginning of the paragraph. “This section presents the empirical evaluation …equation 4 include the regression results for the three types of environmental satisfaction.” have been omitted. The specific modification details are as follows:

This section presents the empirical evaluation of the four hypotheses proposed in this study. Table 4 presents the regression results of the master model (1) where columns 1–3 show the results of regression for the natural ecological environment, hardware facility environment, and daily governance environment, respectively. The explanatory variables for equation 4 include the regression results for the three types of environmental satisfaction.

The regression results for the core variables of equations 1–3 suggest that each of the three types of environment—natural ecological environment, hardware facility environment, and daily governance environment—has a significant positive effect on farmers' well-being at the 1% level of significance, validating H2, H3, and H4, respectively. Hence, higher satisfaction levels with the habitat human settlement environment contribute to higher well-being, validating H1. Table 4 presents the regression results of the master model (1).”

6 – In the Discussion section, the authors begin with the statement, “Attaining well-being is the ultimate goal of all human activities.” While this may be the authors’ opinion, others might disagree, depending on their beliefs, and in any event this statement is extraneous to this study’s hypotheses and findings and may distract from them, so I suggest deleting it.

Later in the same study the authors continue, “The findings of the present study help to reveal the mechanisms that influence farmers’ well-being. Actively exploring ways to enhance the well-being of farmers around nature reserves is important for achieving rural revitalization and ecological civilization in China.” Such statements, slightly revised, would be a much stronger way to start your discussion. The term “ecological civilization” does not quite make sense in English as ecology pertains to nature whereas civilization pertains to human culture. I suggest using a difference phrase to clarify what you mean.

Response: I have revised as suggested.

1-“Attaining well-being is the ultimate goal of all human activities.” had been omitted, the specific modification details are as follows:

 “Attaining well-being is the ultimate goal of all human activities. The findings of the present study help to reveal the mechanisms that influence farmers’ well-being. Actively exploring ways to enhance the well-being of farmers around nature reserves is important for achieving rural revitalization and construction of nature reserve system ecological civilization in China.”

2-“Ecological civilization” was replaced with” construction of nature reserve system” throughout the manuscript. Please refer to the manuscript for details.

7 – In the Discussion the authors also state, “… higher ecological control results in the growth of wildlife populations in nature reserves, which in turn results in wildlife perpetration on the crops and livestock of surrounding villages, leading to increasing human–wildlife conflict. This may explain why in the present study, the regression coefficient for the effect of the wildlife population indicator on well-being was negative. For farmers outside nature reserves, a good ecological environment has led to an increase in wildlife populations. However, such farmers did not experience the distress of wild animals destroying their livestock and crops, which is why wildlife population had a significant positive effect on their well-being.” Is this difference in the implications of growing wildlife populations inside or outside nature reserves due to the abilities of farmers to stop wildlife from destroying their crops, hunt wildlife (and thus control their populations) or something else? A discussion of this difference would be of great interest to wildlife scientists and those concerned with human-wildlife conflict so I suggest adding a paragraph or two to focus on this.

Response: I have revised as suggested. A more detailed explanation is given for the difference regression coefficient of wildlife population indicator on well-being for farmers inside and outside of the nature reserves. The specific modification details are as follows:

“Moreover, higher ecological control results in the growth of wildlife populations in nature reserves, which in turn results in wildlife perpetration on the crops and livestock of surrounding villages, leading to increasing human–wildlife conflict. This may explain why in the present study, the regression coefficient for the effect of the wildlife population indicator on well-being was negative. Human-animal conflicts in nature reserves are common [61]. Nowadays, an excessive number of wild animals in some parts of China have resulted in the saturation of their original management habitats’ capacity of human settlement environment. Therefore, the scope of their activities is continuing to expand, and instances of damage perpetrated by them will occur sporadically. According to incomplete statistics by the provinces in China, 3,811,300 incidents perpetrated by wild animals occurred from 2017 to 2020, causing direct economic losses of over 15.38 billion yuan [38]. As of July 2021, a total of 9 provinces (cities and regions) had issued compensation measures for damage incurred by local terrestrial wildlife. Due to varying natural and economic conditions in different places, the compensation standards fluctuate: Governments of some underdeveloped areas have only limited financial resources and lack special compensation funds for wildlife-induced damage; some governments only provide compensation for the damage caused by the protected animals, excluding the damage to crops caused by other animals [58]. The Chinese government has made progress in relieving the human-wildlife conflicts in nature reserves; however, there are still numerous issues in terms of the wholesomeness of relevant laws and regulations, the compensation standards, and the insurance damage assessments. For farmers outside nature reserves, the increasing number of wild animals indicates a good ecological environment for farmers outside nature reserves. Given that wild animals mainly inhabit remote nature reserves, there are few incidents of wild animals spoiling livestock or destroying crops. Therefore, wild animals have a significant positive impact on the well-being of farmers outside nature reserves. a good ecological environment has led to an increase in wildlife populations. However, such farmers did not experience the distress of wild animals destroying their livestock and crops, which is why wildlife population had a significant positive effect on their well-being.

Thank you so much for all of your comments and suggestions!

Reviewer 3 Report

The paper entitled "Effect of village habitat environment construction around nature reserves on farmers' well-being: a survey based on 1002 farmer households around six nature reserves in China" is a piece of research that presents an exhaustive analysis of farmers' satisfaction with the habitat environment, and daily governance environment. The methodology is based on 1002 farmers' surveys who live within and outside six nature reserves in China. This study exhibits many data and models relevant to science. Mainly, the aim and methods applied are in line with the journal's scopes. The paper contributes to a steamily well-developed investigation in the field of environmental sciences. The results are well organized, and they represent a solid base to analyze the population's well-being. Therefore, I recommend accepting this manuscript after considering some minor suggestions.

The abstract is too descriptive. Author/s could include some statistical results to make it more scientific. This comment could increase the relevance of the paper. Moreover, It would be necessary to consider the importance of the study on a global scale. Even though the topic is crucial for China, what about this topic on a worldwide scale?

In the introduction section, it is necessary to include a location map.

In the methodology, I suggest that the authors include the survey as an annex to this paper.

In the results section, it is essential to incorporate some graphs or maps to visualize the spatial distribution of surveys. Perhaps if the results were presented on several maps, the authors could visualize the spatial differences in well-being.

The results show the interpretation of all surveys. In order to explain the information spatially, it will be necessary to include some maps with triangles and circles indicating the trend and the statistical significance, but considering the different spaces where the authors apply the survey.

In general, there is a lack of spatial information.

Author Response

Response to reviewers #3

Reviewer #3:The paper entitled "Effect of village habitat environment construction around nature reserves on farmers' well-being: a survey based on 1002 farmer households around six nature reserves in China" is a piece of research that presents an exhaustive analysis of farmers' satisfaction with the habitat environment, and daily governance environment. The methodology is based on 1002 farmers' surveys who live within and outside six nature reserves in China. This study exhibits many data and models relevant to science. Mainly, the aim and methods applied are in line with the journal's scopes. The paper contributes to a steamily well-developed investigation in the field of environmental sciences. The results are well organized, and they represent a solid base to analyze the population's well-being. Therefore, I recommend accepting this manuscript after considering some minor suggestions.

The abstract is too descriptive. Author/s could include some statistical results to make it more scientific. This comment could increase the relevance of the paper. Moreover, It would be necessary to consider the importance of the study on a global scale. Even though the topic is crucial for China, what about this topic on a worldwide scale?

Response: Thank you so much for your well-judged comments. I have revised as suggested. 1-Some data are added in the abstract, the specific modification details are as follows:

“Moreover, the satisfaction towards the natural ecological environment, hardware facility environment, and daily governance environment exerts a substantial impact on well-being at the significance level of 1%, with a positive sign, showing a stable enhancement role. Among them, the satisfaction of hardware facility environment is the most essential for improving happiness, with a coefficient of 0.126…Heterogeneity analysis suggests that the positive effect of satisfaction with the human settlement the habitat environment on farmers' well-being within nature reserves is more significant in the natural ecological environment, with a coefficient of 0.244; the hardware facility environment has the greatest positive effect on the well-being of farmers outside nature reserves, with a coefficient of 0.224;… …”

2- The significance of the topic worldwide is placed at the head of the abstract. The red font indicates the modified part, the specific modification details are as follows:

“Numerous countries actively consider human settlement environment and have implemented rural governance strategies to ameliorate the living conditions of rural dwellers.”

In the introduction section, it is necessary to include a location map.

Response: I have revised as suggested. The location map is put in the “Data sources”, as shown in Figure 1.

In the methodology, I suggest that the authors include the survey as an annex to this paper.

Response: The questionnaire belongs to the research group of the National Social Science Foundation.The questionnaire also involves related data and research topics of other team members, so it is not convenient to disclose them.

In the results section, it is essential to incorporate some graphs or maps to visualize the spatial distribution of surveys. Perhaps if the results were presented on several maps, the authors could visualize the spatial differences in well-being.

The results show the interpretation of all surveys. In order to explain the information spatially, it will be necessary to include some maps with triangles and circles indicating the trend and the statistical significance, but considering the different spaces where the authors apply the survey.

In general, there is a lack of spatial information.

Response: I have revised as suggested. The map of average happiness of 44 villages inside and outside the nature reserves is added in the “Variable selection”, as shown in Figure 2.

Figure 2 shows the average happiness of 44 villages inside and outside the nature reserves. In the whole research area, the average happiness index of farmers is 4.00, the average happiness outside nature reserves is 3.96, and the average happiness outside nature reserves is 4.06. There were 16 villages with average happiness greater than 4.00, accounting for 59.3% of all villages outside the nature reserve. And there are 12 villages with average happiness greater than 4.00, accounting for 70.6% of the total villages in the reserve. The five villages with the lowest average happiness were all outside the reserve.

Thank you so much for all of your comments and suggestions!
